Intestinal microbiome and its potential functions in bighead carp (Aristichthys nobilis) under different feeding strategies

Li Xuemei 1
Zhu Yongjiu 1
Ringø Einar 2
Wang Xuge 1
Gong Jinling 1
http://orcid.org/0000-0002-3283-2341 Yang Deguo 1 yangdg@yfi.ac.cn
1 Key Laboratory of Freshwater Biodiversity Conservation, Ministry of Agriculture and Rural Areas of China, Yangtze River Fisheries Research Institute, Chinese Academy of Fishery Sciences , Wuhan , China
2 Norwegian College of Fishery Science, Faculty of Biosciences, Fisheries and Economics, UiT The Arctic University of Norway , Tromsø , Norway
Kormas Konstantinos
Electronic publication date: 2018 Dec 3
Publication date: 2018
Volume: 6
Electronic Location ID: e6000
Received 2018 Jun 7; Accepted 2018 Oct 25
Copyright: © 2018 Li et al.
Copyright year: 2018
Copyright holder: Li et al.
License: This is an open access article distributed under the terms of the Creative Commons Attribution License, which permits unrestricted use, distribution, reproduction and adaptation in any medium and for any purpose provided that it is properly attributed. For attribution, the original author(s), title, publication source (PeerJ) and either DOI or URL of the article must be cited.
License URL: https://creativecommons.org/licenses/by/4.0/

Keywords: Aristichthys nobilis, MiSeq sequencing, Fertiliser, Formulated feed, Intestinal microbiome

Funding: National Natural Science Foundation of China 31502142 Central Public-interest Scientific Institution Basal Research Fund, CAFS 2017JBF0103 China Agriculture Research System (CARS-46) This research was funded by the National Natural Science Foundation of China (31502142), Central Public-interest Scientific Institution Basal Research Fund, CAFS (NO. 2017JBF0103) and the China Agriculture Research System (CARS-46). The funders had no role in study design, data collection and analysis, decision to publish, or preparation of the manuscript.

==============================
Bighead carps (Aristichthys nobilis) were divided into four groups with different feeding strategies: group A, nature live food only (fertiliser only, 200 g urea + 160 g ethylamine phosphate + 250 g Huangjintai bio-fertiliser); group B, nature live food + 1/2 formulated feed; group C, nature live food + formulated feed; and group D, formulated feed only. The intestinal microbiomes of the different groups were compared through the Illumina MiSeq sequencing of the bacterial 16S rRNA gene. The specific growth rate (SGR), survival and blood biochemical factors of the fish were also investigated. Results showed that feeding treatment influenced the intestinal communities in the fish. In specific, more bacterial phyla dominated in groups A and B (phyla Bacteroidetes, Fusobacteria, Firmicutes and Proteobacteria in group A, phyla Proteobacteria and Fusobacteria in group B) than in groups C and D (phylum Proteobacteria). The diversity was also lower in groups C and D than in groups A and B. Unweighted pair-group method analysis revealed a clear difference in intestinal microbiota among the different feeding treatments. No difference in survival rate was found among the treatment groups, but the SGR was significantly higher (P < 0.01) in groups B, C and D than in group A. Functional analysis showed that the intestinal bacteria correlated with fish glucose metabolism in group A but with lipid metabolic activity in groups B, C and D. In summary, the intestinal microbiomes and their potential functions vary in bighead carp under different feeding treatments. This study provides new insights into the gut microbiomes of filter-feeding and formulated diet-fed fish.

Introduction

The gut microbiota of vertebrates, including fish, contributes to nutrition, immunity and development of the host (Ley et al., 2008; Ray, Ghosh & Ringø, 2012; Dinan & Cryan, 2017; Blum, 2017). Approximately 34,000 fish species have been identified (FishBase, 2017), and they are important to understand physiology, ecology and natural history of vertebrates (Wong & Rawls, 2012; Tarnecki et al., 2017). Microorganisms from water and food can adhere and colonise the fish gut, and an imbalanced fish gut microbiota can alter the immune regulatory functions of the gut and contribute to disease manifestation (Pérez et al., 2010; Ghanbari, Kneifel & Domig, 2015). Many factors, such as species, age, developmental stage, geographic location, gender, environmental factors, the individual genetics of fish can modulate the gut microbiota composition (Nayak, 2010; Li et al., 2012, 2013, 2016; Ringø et al., 2016). In addition, diet, including dietary replacement of fishmeal modulate the gut bacterial community in fish (Merrifield et al., 2011; Ye et al., 2014; Baldo et al., 2015; Estruch et al., 2015; Eichmiller et al., 2016; Ringø et al., 2016; Zhou et al., 2017). However, less information is available about differences in the gut microbiomes of fish species fed live food or artificial feed (Savas, Kubilay & Basmaz, 2005; Ni et al., 2014). Ni et al. (2014) revealed that the gut microbiota of grass carp (Ctenopharyngodon idella) was modulated by ryegrass (Lolium perenne) or commercial feed, and this modulation help to digest otherwise undigested dietary polysaccharose to achieve nutritional and physiological homeostasis.

In China, bighead carp (Aristichthys nobilis), silver carp (Hypophthalmichthys molitrix), grass carp and black carp (Mylopharyngodon piceus) are the major carp species (Yu, Tang & Li, 2010). Following the successful breeding of these species during the late 1950s (Zhong, Li & Zhang, 1965), bighead carp has become one of the most intensively exploited fish species in aquaculture, with an annual global production of over 10 million tons in 2015 and China is the main producer (Food and Agriculture Organization of the United Nations (FAO), 2015). As filter feeders, bighead carp preferentially consumes zooplankton, but also ingests phytoplankton and detritus, and they are also used as a potential biological control agent to improve water quality and increase fish production in culture ponds (Lazareva, Omarov & Lezina, 1977; Burke, Bayne & Rea, 1986; Xie & Liu, 2001; Conover, Simmonds & Whalen, 2007). With the increasing demand of bighead carp for consumption, pond models with a high density of carp has received considerable attention (Mi, Wen & Ge, 2016). Moreover, feeding bighead carp formulated feed to increase farm yield has become popular and acceptable (Huang & Pan, 2013; Mi, Wen & Ge, 2016). Considering the influence of diet on fish gut bacterial community, it is of interest to extend the knowledge of bighead carp gut microbiome and their potential metabolic function when the fish is fed natural live food or formulated feed.

The aim of this study was evaluate (1) how the intestinal microbiome structure is modulated by different feeding regimes, filter-feeding and formulated diet; (2) and what’s the relationship between intestinal microbiome and metabolic functions of bighead carp. The results will improve our understanding of the gut microbiome of filter-feeding and feeding formulated diet fish.

Materials and Methods

Experimental designs

The experiment was conducted in 12 rectangular enclosure (length 9 m × width 6 m × height 2.5 m) in earth ponds (2666.4 m2) at Yaowan fish farming base (30.16N, 112.18E) in Yangtze River Fisheries Research Institute in Jingzhou City, China. Four different pond management systems, triplicate ponds, were used. Group A (nature live food only): fertilised ponds to obtain zooplankton; cladocera, copepod and rotifer, which were identified in lab according to Jiang & Du (1979), Shen (1979) and Wang (1961), group B (nature live food +1/2 formulated feed): fertilised ponds in which bighead carp were fed 1–1.5% of the body weight (BW) with formulated feed, group C (nature live food + formulated feed): fertilised ponds where fish were fed 2–3% of the BW with formulated feed and group D (only formulated feed): fish were fed 2–3% of the BW with formulated feed.

Pond preparation and management

All ponds were drained, renovated and sterilised with Lime (CaO) prior to the experiment. Each pond was filled with well water to 50 cm and treated with compound fertiliser (200 g urea + 160 g ethylamine phosphate + 250 g Huangjintai bio-fertiliser (made of fish protein, dairy products, astragalan and functional peptides, Hubei Daming Aquatic Science and Technology Co., Ltd, Jingzhou, Hubei, China) for 1 week before experimental start. This was done to culture natural live food for bighead carp. Thereafter, the water level was increased up to 160 cm. The fertiliser was used twice every week during the experiment in pond A, B and C. Commercially formulated feed (Zhengchang Company, Changzhou, Jiangsu, China; diameter: 4.0–5.0 mm) was fed to fish in pond B, C and D. The biochemical composition of the formulated feed was; crude protein ≥34.0%, crude lipid ≥3.0%, lysine ≥1.4%, total phosphorus ≥1.0%, crude ash ≤15.0%, crude fiber ≤12.0%, calcium = 1.0–4.0% and moisture ≤ 12.0%.

Fish were transferred from Hubei Daming Aquatic Science and Technology Co., Ltd in Jingzhou City to the Yaowan fish farming base. A total of 192 fish with initial BW of 906.7 ± 102.4 g and body length (BL) of 37.6 ± 2.0 cm were randomly distributed to the ponds, 16 fish per pond, where they were fed formulated diet at a rate of 2–3% of biomass twice a day (9:00 and 16:00). Each pond was equipped with one nanodisk to ensure adequate oxygen level. The experiment was carried out from April 2014 to September 2014, and the main environmental factors of the ponds are displayed in Table S1.

Sample collection and pre-processing

A total of 180 days after the experimental start, fish were captured with falling nets in order to avoid additional stress responses. The falling nets were used twice in each pond, and one or two fish from each pond were randomly collected and anesthetised with an overdose (70 mg/L) of MS 222 (Syndel, Ferndale, WA, USA). Final BL, BW and whole length were measured prior to blood sampling, and specific growth rate (SGR) (% d−1) was calculated: SGR = [(ln final weight−ln initial weight)/rearing duration in days] × 100. Blood was collected from caudal artery by sterile syringes and transferred into sterile tubes and centrifuged at 3.000 rpm for 10 min at 4 °C. The separated serum was transported to the laboratory under refrigeration and stored at −80 °C prior to biochemical analysis.

Fish exterior surfaces were swabbed with 75% ethanol before the ventral midline was dissected. Faecal content was collected using sterile scalpel and forceps into a sterile tube by squeezing along the exterior side of the intestine as described elsewhere (Li et al., 2014; Ye et al., 2014). Intestinal samples of 23 fish (group A, seven fish; group B, seven fish; group C, five fish; and group D, four fish) were immediately frozen in liquid nitrogen, transported to the laboratory and then stored at −80 °C until DNA extraction. All samples were collected within 1 h post-fish capture.

The experiments were performed in accordance with the Regulations for the Administration of Affairs Concerning Experimental Animals of China. The protocols applied in the present study were approved by the Institutional Animal Care and Use Committee of the Yangtze River Fisheries Research Institute, Chinese Academy of Fishery Sciences (Approval ID: CAFSCJ-2014-001).

Blood biochemical parameters

Frozen blood samples were first thawed at −20 °C and then at 4 °C as described by Zhang et al. (2010). Blood biochemical parameters, alanine aminotransferase (ALT), aspartate aminotransferase (AST), alkaline phosphatase (ALP), total protein (TP), glucose (GLU), triglyceride (TG), total cholesterol (TC), high-density lipoprotein cholesterol (HDL-C) and low-density lipoprotein cholesterol (LDL-C), were analysed with an Olympus® AU2700 Automated Chemistry Analyzer using commercial kits (D-20097; Olympus life and Material Science Europa GmbH, Hamburg, Germany) at Hubei Provincial Hospital of TCM.

DNA extraction, PCR and sequencing

For the analysis of bacterial diversity, 0.25 g (wet weight) of the intestinal samples was used to extract DNA by the Powerfecal DNA Isolation kit (Mo Bio Laboratories Inc., Carlsbad, CA, USA) in accordance with the manufacturer’s protocols. The 338F (ACTCCTAC GGGAGGCAGCA) and 806R (GGACTACNNGGGTWTCTAAT) primers were used to amplify the bacterial 16S rRNA gene V3–V4 fragments. PCR integrant and protocols were carried out as described by Gu et al. (2016): 95 °C for 2 min, followed by 27 cycles at 95 °C for 30 s, 55 °C for 30 s, and 72 °C for 45 s and a final extension at 72 °C for 10 min, 10 °C until halted by user.

The PCR products were separated by 2% agarose gel electrophoresis and negative controls were always performed to make sure there was no contamination. All bands of the desired size (approximately 468 bp) were purified using the AxyPrep DNA Gel Extraction Kit (Axygen Biosciences, Union City, CA, USA). Prior to sequencing, purified PCR products were quantified by Qubit®3.0 (Life Invitrogen, Waltham, MA, USA) and every 24 amplicons whose barcodes were different were mixed equally. The pooled DNA product was used to construct Illumina Pair-End library following Illumina’s genomic DNA library preparation procedure. Then the amplicon library was paired-end sequenced (2 × 250) on an Illumina MiSeq platform (Shanghai Majorbio Bio-Pharm Technology and Lingen Biotechnology Co., Ltd) according to the standard protocols.

Process of sequencing data

Trimmomatic and QIIME (version 1.17) was used to process and quality-filter the raw fastq files (Caporaso et al., 2010; Gu et al., 2016). Three criteria were followed: (i) reads were truncated at any site receiving an average quality score <20 over a 50 bp sliding window, discarding the truncated reads that were shorter than 50 bp; (ii) Exact barcode matching, <20% mismatches were allowed, and reads containing ambiguous characters were removed; (iii) only sequences that overlap by longer than 10 bp were assembled according to their overlap sequence; (iv) adjust the sequence direction, the mismatch number of barcode is 0, and the maximum primer mismatch number is 2 (Sun et al., 2015). UPARSE was used to cluster operational taxonomic units (OTUs) with 97% similarity cutoff, and UCHIME was applied to identify and remove chimeric sequences based on both mode reference database and de novo. The phylogenetic affiliation analysis of each 16S rRNA gene sequence was introduced by RDP Classifier against the SILVA (SSU115)16S rRNA database with a confidence threshold of 70% (Schloss & Westcott, 2011; Westcott & Schloss, 2015).

Statistical analysis

Rarefaction analysis based on treatment and technical replicates was performed after sequence re-sampling using the Mothur program (version 1.30.1, http://www.mothur.org/wiki/Schloss_SOP#Alpha_diversity). Alpha diversity indices were determined from rarefied tables using the Shannon–Wiener index and Simpson index for species diversity and the Chao1 index for species richness to reveal changes in intestinal microbiota in different samples (Caporaso et al., 2011). The unweighted pair-group method based on Bray–Curtis dissimilarity was used to perform a hierarchical clustering of different samples. Taxonomic composition-based non-metric multidimensional scaling analysis and weighted UniFrac distance-based PCoA analysis were conducted to illustrate the overall patterns of microbial communities in the different samples. Multiple regression of environmental variables with the microbial community groups was analysed. Independent regression models of genus taxonomy and biochemical parameters were established to screen the microbial genera that could significantly predict metabolic characters and to explore the potential relationships between intestinal microbes and host metabolism. Regression analysis was run on the entire dataset, and only significant differences were shown. Moreover, functional predictions on family-level microbiome were also performed in PICRUSt. All data were expressed as mean ± SD. Two-tailed Student’s t-test was used to assess fish growth parameters and metabolic differences, and false discovery rate correction (Benjamini–Hochberg) was considered. Multivariate ANOVA was used to assess the differences in bighead carp intestinal bacterial communities between the different treatments. Statistical analyses were performed with the software SPSS 22.0 (IBM, New York, NY, USA) and R (ver. 3.0.1) package (R Core Team, 2013). The level of significance was set at a P-value of < 0.05.

Results

Growth performance and biochemical parameters

Feed application significantly affected fish growth and SGR. The SGRs of bighead carp were significantly higher (P < 0.01) in groups B, C and D than that in group A, whereas the survival rate showed no difference among the treatments (Table 1).

Table 1 Main growth performance of bighead carp in different treatments.

	Group A	Group B	Group C	Group D	P-value	
Final weight (g)	965.8 ± 125.3a	1233.6 ± 343.9b,c	1188.3 ± 284.4b,c	1426.6 ± 159.0c	<0.01	
Survival (%)	96.0 ± 5.20%	70.8 ± 30.2%	75.0 ± 10.5%	70.8 ± 10.5%	NS	
SGR (% d−1)	0.04 ± 0.03a	0.17 ± 0.16b	0.16 ± 0.08b	0.25 ± 0.06b	<0.01	
Note:

Mean ± SD.

a,b,c Indicates significant association (P < 0.05).

Biochemical blood parameters, mean and SD are shown in Table 2. The concentrations of ALT and LDL-C were significantly lower (P < 0.01) in groups A, B and C than in group D. TP, TC, TG, GLU and HDL-C were significantly (P < 0.01) lower in group A compared to the other groups. No significant (P > 0.05) differences in ALP and AST levels were noticed among the different treatments.

Table 2 Comparison of metabolic differences between fish groups under different treatments.

	Group A	Group B	Group C	Group D	P-value	
ALT(U/L)	23.6 ± 3.90a	29.1 ± 8.90a	33.8 ± 13.3a	71.3 ± 13.3b	<0.01	
AST(U/L)	57.6 ± 21.2	36.7 ± 5.20	45.2 ± 10.5	42.5 ± 4.70	NS	
ALP(U/L)	32.0 ± 19.3	69.0 ± 35.7	70.4 ± 25.1	61.3 ± 38.8	NS	
TP (g/L)	22.4 ± 2.70a	29.7 ± 4.50b	29.5 ± 1.90b	31.2 ± 3.50b	<0.01	
TC (mmol/L)	1.80 ± 0.40a	2.60 ± 0.50b	2.70 ± 0.40b	2.80 ± 0.10b	<0.01	
TG (mmol/L)	0.40 ± 0.10a	2.20 ± 0.40b	1.70 ± 0.20c	1.70 ± 0.10c	<0.01	
GLU (mmol/L)	4.50 ± 0.40a	5.30 ± 1.10a,b	6.10 ± 1.10b	7.90 ± 0.80c	<0.01	
HDL-C (mmol/L)	0.20 ± 0.10a	0.30 ± 0.10b	0.30 ± 0.10b,c	0.4 ± 0.10c	<0.01	
LDL-C (mmol/L)	0.50 ± 0.20a	0.60 ± 0.10a,b	0.06 ± 0.20a,b	0.80 ± 0.10b	<0.05	
Notes:

Mean ± SD.

ALT, alanine transaminase; AST, aspartate aminotransferase; ALP, alkaline phosphatase; TP, total protein; TC, total cholesterol; TG, triglyceride; GLU, glucose; HDL-C, high-density lipoprotein; LDL-C, low-density lipoprotein.

a,b,c Indicates significant association (P < 0.05).

Intestinal microbiota diversity and richness

After quality filtering and length trimming, 759,048 high-quality bacterial sequences were obtained, equivalent to an average of 33,002 (min 26,755 and max 38,788) reads per sample, when representative OTU sequences were classified using the RDP classifier.

The number of OTUs was analysed for each sample with a 97% sequence similarity cut off value. Alpha diversity metrics showed no significant (P > 0.05) differences in OTU richness (Chao1 index) among the treatments (Fig. 1). Meanwhile, the Shannon–Wiener and Simpson indices significantly differed (P < 0.05) among the feeding strategies (Fig. 2). Group A and B had the highest diversity; significantly (P < 0.05) different from groups C and D, By contrast, the diversities between groups A and B or between groups C and D revealed no significant (P > 0.05) difference.

Figure 1 Rarefaction analysis of MiSeq sequencing reads of the 16S rRNA gene in different fish samples with different treatments.

Rarefaction curves at a cutoff level of 3% were constructed at a 97% sequence similarity cutoff value. A, fertiliser; B, fertiliser + 1/2 feeding; C, fertiliser + feeding; D, feeding.

Figure 2 Alpha diversity Shannon (A) and Simpson (B) measures based on average operational taxonomic units (OTUs) of fish with different treatments.

Error bars indicate SD. a, b indicate significant association (P < 0.05). A, fertiliser; B, fertiliser + 1/2 feeding; C, fertiliser + feeding; D, feeding.

Intestinal microbiota composition

Phyla Bacteroidetes, Fusobacteria, Firmicutes and Proteobacteria were dominant in group A (Fig. 3A). In group B, phyla Proteobacteria and Fusobacteria dominated the intestinal microbiome and constituted of 60.0% ± 25.3% and 18.5% ± 20.4%, respectively, followed by Cyanobacteria, Bacteroidetes and Firmicutes. Phylum Proteobacteria was dominant in the intestinal microbiomes of group C (96.1% ± 2.5%) and group D (94.5% ± 6.3%), whereas other phyla comprised < 2% of the total reads.

Figure 3 Distribution of average read number among the major phyla (A) and major class (B) in fish intestinal microbiota with different treatments.

A, fertiliser; B, fertiliser + 1/2 feeding; C, fertiliser + feeding; D, feeding.

Figure 3B revealed that family Porphyromonadaceae (40.2 ± 22.3%), Fusobacteriaceae (29.7 ± 23.3%) and Peptostreptococcaceae (12.2 ± 7.6%) dominated the intestinal microbiome of group A. Family Gammaproteobacteria_unclassified OTU (31.6 ± 26.7%), Fusobacteriaceae (18.8 ± 27.8%), Aeromonadaceae (14.6 ± 13.5%) and Rhodocyclaceae (4.85 ± 4.70%) dominated the intestinal microbiome in group B. In group C and D, family Gammaproteobacteria_unclassified OTU were the dominant intestinal microbiome, with a portion of 85.3 ± 11.7% and 83.0 ± 17.0% of total reads, respectively. At the genus level, significant (P < 0.01) differences were revealed among the treatments. The abundance of Cetobacterium (phylum Fusobacteria, family Fusobacteriaceae), Peptostreptococcaceae_incertae_sedis OTU (phylum Firmicutes, family Peptostreptococcaceae) and Porphyromonadaceae_uncultured OTU (phylum Bacteroidetes, family Porphyromonadaceae) were significantly (P < 0.05) higher in group A when compared to the three other groups. The genera Gammaproteobacteria_unclassified OTU (phylum Proteobacteria), Aeromonas and Pseudomonas (phylum Proteobacteria, family Aeromonadaceae and Pseudomonadaceae, respectively) and the genus Cetobacterium were present at higher proportions in group B than in the other groups. The abundance of the genus Aeromonas was significantly (P < 0.01) higher in group B than in group A, C and D. By contrast, the genus Gammaproteobacteria_unclassified OTU was significantly higher (P < 0.01) in group C and D than in the other groups (Table 3). Meanwhile, the shared taxa with relative abundance above 1% were further examined to evaluate core bacterial shifts among different treatments. Clear core bacterial turnover patterns among different treatments were visualised by the heat maps, and no individual OTUs were shared across all diet combination treatments (Fig. S2).

Table 3 Average relative abundances (% of sequences per treatment) and standard deviation of the most abundant bacteria at the genus taxonomy level in fish intestine.

Phylum	Family	Genus	Group A (%)	Group B (%)	Group C (%)	Group D (%)	P-value	
Proteobacteria	Aeromonadaceae	Aeromonas	0.20 ± 0.24a	20.7 ± 12.63b	1.38 ± 1.10a	0.98 ± 1.23a	<0.01	
Proteobacteria		Gammaproteobacteria_unclassified OTU	0.95 ± 0.76a	50.3 ± 38.9b	95.8 ± 2.82c	93.8 ± 6.31c	<0.01	
Proteobacteria	Pseudomonadaceae	Pseudomonas	0.02 ± 0.01a	4.39 ± 3.02b	1.93 ± 1.62b	2.78 ± 1.22b	<0.05	
Fusobacteria	Fusobacteriaceae	Cetobacterium	34.5 ± 27.3a	21.8 ± 4.12a	0.72 ± 0.79b	1.29 ± 0.86b	<0.05	
Firmicutes	Peptostreptococcaceae	Peptostreptococcaceae incertae_sedis OTU	16.3 ± 11.9a	0.19 ± 0.17b	0.08 ± 0.04b	0.04 ± 0.03b	<0.01	
Bacteroidetes	Porphyromonadaceae	Porphyromonadaceae_uncultured OTU	48.1 ± 24.9a	2.61 ± 1.82b	0.17 ± 0.17b	1.06 ± 0.20b	<0.01	
Notes:

Mean% ± SD.

a,b,c Indicates significant association (P < 0.05).

Intestinal microbiota community composition

Hierarchical clustering showed that bacterial communities clustered as a consequence of feeding strategy treatments (Fig. 4). The microbiota community of group B dispersed: some clustered with group A, while others clustered with groups C and D. A higher separation was revealed between group A compared with groups C and D, as the two latter groups generally clustered together.

Figure 4 Unweighted pair-group method dendrograms showing the similarity of fish intestinal microbiota with different treatments based on operational taxonomic units (OTUs).

A, fertiliser; B, fertiliser + 1/2 feeding; C, fertiliser + feeding; D, feeding.

Functional analysis

The relationships between genera and biochemical parameters were investigated separately through independent regression models to explore the potential metabolic functions of the intestinal microbiome in bighead carp. Genera Gammaproteobacteria_unclassified OTU, Pseudomonas, Cetobacterium and Porphyromonadaceae_uncultured OTU were significantly (P < 0.05) related to the fish biochemical parameters (Fig. 5). Results showed that ALT and GLU were positively associated with the genus Gammaproteobacteria_unclassified_OTU (R2 = 0.27 and R2 = 0.46, respectively). By contrast, GLU was negatively associated with the genera Cetobacterium and Porphyromonadaceae_uncultured OTU (R2 = 0.24 and R2 = 0.27, respectively). Moreover, AST was negatively correlated with the genus Pseudomonas, whereas TG was positively correlated with the genus Pseudomonas (R2 = 0.72 and R2 = 0.79, respectively).

Figure 5 Scatter diagram (A–E) depicting the linear relationships of genus taxonomy and metabolic factors (all P-values < 0.05).

ALT, alanine transaminase; GLU, glucose; AST, aspartate aminotransferase; and TG, triglyceride.

Discussion

Recently in China, the filter-feeding fish bighead carp has been successfully fed formulated feed to increase the farm yield to meet the increasing demand (Mi, Wen & Ge, 2016). In the present study, the SGR of group D was significantly (P < 0.05) higher than that in group A, and our results are in accordance with Mi, Wen & Ge (2016), suggesting feeding formulated feed could improve the growth rate of bighead carp. Though the survival rate is no significantly different between all treatments, 20–25% reduced survival existed in the group B, C and D, it may due to the poor condition of fish and lower dissolved oxygen in rainy day. In a previous study, Asadi et al. (2006) reported that ALT and AST are mainly located in the liver and reflect its physiological state. The significant (P < 0.05) increase in ALT activity in group D indicated a higher activity in the amino acid catabolism of the liver. The low GLU and TG in group A may be due to the placid behaviour of bighead carp under natural conditions, as bighead carp is more active to catch the formulated feed in groups B, C and D than in group A (Song & Kong, 2013). These differences in catching food may induce more glycogen and protein catabolism, which affected the concentrations of GLU and TP (Atencio, Edwards & Pesti, 2005).

The gut microbial community of fish is modulated by dietary manipulations (Muegge et al., 2011; Wu et al., 2011; Ringø et al., 2016). However, to our knowledge, information about the intestinal microbiome of filter-feeding fish fed formulated feed is lacking. The results of present study improve the knowledge on the microbial communities of filter-feeding fish and feeding fish, and might be exploited in formulated feed production in the future. In general, fish intestinal microbiota is dominated mainly by the phyla Proteobacteria and Firmicutes (Navarrete et al., 2010; Sullam et al., 2012; Estruch et al., 2015; Miyake, Ngugi & Stingl, 2015), while Fusobacteria was the dominant phylum in the current study, a finding which is in accordance with that revealed for common carp (Cyprinus carpio L.) (Van Kessel et al., 2011). Bacteroidetes is an abundant phyla in bighead carp (the present study), silver carp (Ye et al., 2014), paddle fish (Psephurus glades) (Li et al., 2014), sea bass (Dicentrarchus labrax) (Carda-Diéguez, Mira & Fouz, 2014) and marine herbivorous fish (Sullam et al., 2012). Interestingly, the core intestinal microbial composition of bighead carp in group A was more consistent with that previously reported in paddle fish that similar dominant phyla Bacteroidetes, Fusobacteria, Firmicutes and Proteobacteria were revealed (Li et al., 2014). This result may due to the fact that natural food consumed by paddle fish (Zhu, Li & Yang, 2014) is similar to that consumed by bighead carp in the present study.

In the present study, the intestinal microbial composition and community of bighead carp had no relationships with the environmental variables (Table S2). However, they were significantly influenced by the formulated feed; that is, bacterial community was unique for groups A, C and D, whereas that for group B was variable (Fig. 3; Fig. S1). Considering that more bacterial species and higher diversity of intestinal microbiome were revealed in groups A and B than in other groups, we hypothesised that formulated feed reduces the species and diversity of intestinal microbiome in bighead carp. The variation in eaten natural food and its associated microbes may influence the gut bacteria diversity because bighead carp is reportedly predominantly zooplanktivorous, and the fish may eat phytoplankton and detritus when the concentrations of zooplankton are low (Zhang, Xie & Huang, 2008). However, Bolnick et al. (2014) elucidated that multiple diet components can interact non-additively to modulate the gut microbial diversity in three spine stickleback (Gasterosteus aculeatus) and Eurasian perch (Perca fluviatilis). Moreover, despite the decreased diversity of intestinal microbiome, the growth performance was improved in groups C and D. However, previous study has reported that pig’s BW was significantly decreased when feeding deoxynivalenol contaminated wheat, while there were no significant difference in their intestinal bacterial diversity comparing with control group (Li et al., 2017).Whether fish growth performance is related to intestinal microbial diversity merits further investigations.

In this study, genus Gammaproteobacteria_unclassified OTU substantially increased in groups B, C and D and positively correlated with GLU and ALT concentrations, indicating that this genus may be positively associated with carbohydrate and lipid metabolism. The contribution of gastrointestinal microbiota to host carbohydrate and lipid metabolism has been intensively studied in human, mice, cow and grass carp (Turnbaugh et al., 2006; Brulc et al., 2009; Velagapudi et al., 2010; Ni et al., 2014).

Previous studies have reported that Cetobacterium somerae is a common and widely distributed species within the guts of freshwater fishes, and its prevalence is negatively correlated with the dietary availability of vitamin B12 (Tsuchiya, Sakata & Sugita, 2008; Eichmiller et al., 2016). Hence, Cetobacterium somerae has been assumed to have a main role in the synthesis of vitamin B12 in the fish gut (Sugita & Miyajima, 1991). However, as vitamin B12 within the natural food was not investigated in the current study, further studies on this topic are needed. Supplementation of vitamin B12 in formulated fish feed may lower the abundance of Cetobacterium in groups C and D. However, to verify this controversial hypothesis further studies are needed. In mice and human, Porphyromonadaceae negatively affects lipid metabolism, and it is associated with non-alcoholic fatty liver disease, atherosclerosis and diabetes in human (Henao-Mejia et al., 2012; Marques et al., 2015), while Peptostreptococcaceae were revealed to be positively correlated with lipid metabolism in bighead carp (Fig. S3). In addition, Cetobacterium and Porphyromonadaceae_uncultured OTU, as the dominant genera in group A, were revealed to be negatively associated with GLU concentrations (Fig. 5), suggesting the fish GLU metabolism in the fertiliser group may be limited by these bacteria. However, Fusobacteriaceae and Peptostreptococcaceae both showed positive correlation with carbohydrate metabolism (Fig. S3), more metagenomic sequencing and functional activity study of intestinal microbiome in bighead carp are needed in the future.

In the current study, the genus Pseudomonas was negatively correlated with AST activity but positively correlated with TG, suggests that Pseudomonas may be positively associated with lipid metabolism. Family Pseudomonadaceae also showed a positive correlation with lipid metabolism (Fig. S3). Considering that genus Gammaproteobacteria_unclassified OTU and Pseudomonas were highly dominated in groups B, C and D and both bacteria were positively related to lipid metabolism, we assumed that intestinal bacteria could enhance lipid metabolic activity for bighead carp fed formulated feed.

Conclusions

In summary, the remarkable effect of feeding strategies on the intestinal microbiota of bighead carp highlight the need to determine how different feeding strategies modulate the intestinal microbiota and how this modulation affect the host. Higher bacterial diversities were shown in group A and B than in group C and D. The core intestinal microbiome in group A comprised the phyla Bacteroidetes, Fusobacteria, Firmicutes and Proteobacteria, whereas phyla Proteobacteria and Fusobacteria dominated in group B and only phylum Proteobacteria in groups C and D. Basing on the relationships between intestinal microbiome and the metabolic functions revealed in the present study, we suggest that limited carbohydrate metabolism is presented in group A, while high lipid metabolic activity exists in groups B, C and D. However, the regulatory mechanisms of intestinal microbiome on the metabolism of bighead carp using other techniques and the suitable feed formula for bighead carp based on intestinal microbiota functions require further elucidation.

Supplemental Information

Supplemental Information 1 Figure S1. Non-metric multidimensional scaling analysis and principal coordinate analysis (PCoA) based on based on the taxonomic composition (left) and weighted UniFrac distances (right) of fish intestinal microbiota.

A: fertiliser; B: fertiliser+1/2 feeding; C: fertiliser+ feeding; D: feeding.

Click here for additional data file.

Supplemental Information 2 Figure S2. Heat map showing the dominant genera of intestinal microbiota in bighead carp with different treatments.

A: fertiliser; B: fertiliser + 1/2 feeding; C: fertiliser + feeding; D: feeding.

Click here for additional data file.

Supplemental Information 3 Figure S3. Heat map showing the correlation of family-level intestinal bacteria with their function at L2 level after blasting to KEGG database.

Click here for additional data file.

Supplemental Information 4 Supplemental.

Table S1. Main environmental factors of pond with different treatments. Mean±SD.

Table S2. Relationships of environmental variables with the microbial community groups (Pearson coefficient).

Click here for additional data file.

We thank the graduate students Degao Xu, Fei Li, Haocheng Li and Jianwei Yao for their help during sample collection and Yaowan fish farming for supporting this study.

Additional Information and Declarations

Competing Interests

Author Contributions

Animal Ethics

Data Availability

The authors declare that they have no competing interests.

Xuemei Li conceived and designed the experiments, performed the experiments, analysed the data, prepared figures and/or tables, authored or reviewed drafts of the paper, approved the final draft.

Yongjiu Zhu conceived and designed the experiments, contributed reagents/materials/analysis tools, authored or reviewed drafts of the paper, approved the final draft.

Einar Ringø conceived and designed the experiments, prepared figures and/or tables, authored or reviewed drafts of the paper, approved the final draft, English Language Editing.

Xuge Wang performed the experiments, analysed the data, contributed reagents/materials/analysis tools, authored or reviewed drafts of the paper, approved the final draft.

Jinling Gong performed the experiments, analysed the data, contributed reagents/materials/analysis tools, authored or reviewed drafts of the paper, approved the final draft.

Deguo Yang conceived and designed the experiments, authored or reviewed drafts of the paper, approved the final draft.

The following information was supplied relating to ethical approvals (i.e. approving body and any reference numbers):

The experiments were performed in accordance with the Regulations for the Administration of Affairs Concerning Experimental Animals of China. The protocols applied in the present study were approved by the Institutional Animal Care and Use Committee of the Yangtze River Fisheries Research Institute, Chinese Academy of Fishery Sciences (Approval ID: CAFSCJ-2014-001).

The following information was supplied regarding data availability:

NCBI Sequence Read Archive: accession number SRS2374735.

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
