# Peer review of "Intestinal microbiome and its potential functions in bighead carp (Aristichthys nobilis) under different feeding strategies"

_PeerJ, doi:10.7717/peerj.6000_

## Round 0.1 · original submission · Major Revisions

Please provide a detailed point-by-point to all of the reviewers suggestions and comments.

Reviewer 1 ·

Basic reporting

The paper is written in clear and unambiguous English, except in few places as it is pointed out in the comments to the author.

The introduction is pretty poor and has to be improved substantially. The main drawback is insufficient background/context of the study.

Professional quality figures, tables provided.

Experimental design

The experimental design is simple and the experiment is well executed to answer the research question.

Some parts of the methods are pretty vague and not well described thus leading to confusion. Specific parts are pointed out in the comments to the author.

Validity of the findings

Though the dataset reveals interesting patterns in the microbial community composition based on the feeding type, the sample number used in the study is too few (n=4-6) compared to the contemporary standards. This might have had an effect on the conclusions.

Additional comments

The article describes the differences in the intestinal microbial communities of the big head carp fed naturally (with zooplankton) and the formulated diets. In addition, authors have also presented the biochemical properties of the blood in relation to the feeding regime. In conclusion, the authors claim that the formulated diet has an impact on the microbial populations as well as certain biochemical parameters of the blood.
The article is generally well written and easy to read, though I have the following concerns.

Major issues

1. The feeding groups could be renamed based on the feeding type (as written in the title). It’s a bit confusing to a reader that the paper starts with the feeding regime experiment and then when it comes to naming the groups, they are named based on the pond fertilization strategy.
2. The introduction is not cohesive and does not lead to the objective in question. To make it interesting for the reader I would strongly recommend that the introduction should be modified/rewritten by bringing in specific examples of diet affecting the microbiome (either from other fish/mammalian experiments).
3. The amplicon generation protocol is not clear in either the manuscript or the provided reference, Gu et al 2016. In addition, the Gu et al 2016 has an error in their manuscript. They say that primers “ 27F 5’-barcode- AGAGTTTGATCCTGGCTCAG-3’ and 533R 5’-AGAGTTTGATCCTGGCTCAG -3’” was used for the amplification and in the next paragraph it is mentioned that “The PCR products were separated by 2% agarose gel electrophoresis and bands of the desired size (approximately 250 bp) were purified”. Theoretically, 27F and 533R primers do not produce amplicons with length 250. The amplicon generation protocol should be clearly elaborated (construction of the amplicon primers, single index, dual index details should be provided) or a proper reference should be included.
4. Line 136, “(i) 200 bp reads”- Is it 200 bp or 250 bp. Sequencing length is 250 bp in the previous part of the manuscript. If the forward and reverse reads are trimmed to 200 bp then I do not think that they can be assembled.
5. Figure 1 is a plot showing the rarefaction. The difference in the mean CHAO1 index and the statistical differences are not depicted in the plot. hence a plot like FIG2 should be made for chao1 index as well.
6. Line 270 “Whether growth performance is related to intestinal microbial
diversity merits further investigations”. This is a key finding of the study. It has to be elaborated in relation to the other farmed animal studies/references that are raised using formulated diets. Perhaps less diversity of bacteria and better growth of the animal go hand in hand?

Minor issues

1. Line 69, “rectangular enclosure ecosystem”- the word ecosystem should be deleted.
2. Line 73, were the mentioned zooplankton species isolated and identified? It should be described in the methods.
3. Line 74, What do you mean by 1/2 feeding? I assume it is feeding half of the recommended feeding rate?. If yes please mention it clearly by introducing a line of text in the methods.
4. Line 99, the exact dose used for the killing should be clearly mentioned.
5. Line 117, the short forms must be written in full.
6. Lines 153 and 156, the words treatments should be replaced by samples.
7. There are some unwanted characters above all the scatter plots that should be removed.
8. Table 1- the survival rates- Though not significantly different, what could be the reason for ~20-25% reduced survival in the group B, C, D?. This has to be put into context in the discussion.
9. It is not clear what the authors mean by “main performance” in the table 1 header ?. Should be replaced with a more understandable word.
10. Line 114, “Approval ID: CAFSCJ-2014-001” The document in the supplementary materials should be cited for this statement.

Reviewer 2 ·

Basic reporting

The paper is well written and clear. There are some language mistakes and some unclear sections that should be improved. More specifically, see here:

Line 40: important to understand these things in what? vertebrates? among diverse fish groups?
Line 48-49: maybe say why knowing these things matter.
Line 63: 2) "and to determine the relationship" instead of "and what's"
Line 82: Not clear what this means: "This was done to get natural fish food organisms."
Line 91: add "to" so that it reads "randomly distributed to the ponds"
Line 97: should read "One hundred and eighty days after the start of the experiment,"
Line 127-133: It is unclear of the authors used proper negative and/or kit controls for sequencing. Especially important since they handle the samples so much, like running on 2% agarose gel and then gel-purifying the products etc.

Line 221-223: I do not believe these are accurate names for genera. It may be better to refer to these sequences as OTUs. For example, Gammaproteobacteria_unclassified is not a genus.

Line 243: add "the" so that it reads "the results of the present study"
Line 293: change "indicating" to "suggests"
Line 301: remove comma after "carp"
Line 305-6: change "basing from" to "Based on"

Experimental design

Experimental design is clear and for the most part done okay. However, all the arguments seem to be made at the phylum level, e.g., 241-256, yet the arthors are making predictions about function. It would be very helpful if the authors made arguments based on family-level predictions, which is the level for which functional constraints or predictions are made. For example, Fig. 3, would be more meaningful if compared at the family level.. The authors try to discuss, quite a bit, of the genera involved, however, I think it would make more sense to describe families of relevant bacteria throughout and make arguments about function then. The work on functional predictions would be substantially improved if they performed analysis such as PICRUSt or something similar.

Validity of the findings

Again, the findings would be significantly strengthened by also comparing familly-level predictions to modeled findings in PICRUSt

---

## Round 0.2 · Minor Revisions

As you can see, there is only one minor issue that needs to handled by you. I believe you will be able to address it very soon. Please provide with a detialed response to this comment along with your revised manuscript.

Reviewer 1 ·

Basic reporting

Introduction improved based on the comments from last round of review

Experimental design

Experimental design has more clarity compared to the first version.

Validity of the findings

Better explained in the revised manuscript.

Additional comments

The manuscript has been improved considering the comments from the last round of review, although there are few things that needs to be addressed.

The authors wrote in the rebuttal that only the PCR recipe and the thermocycling conditions were used from the Gu et al. Hence I request the authors to write in detail the protocol that they used for the preparation of the libraries for the illumina sequencing.

Reviewer 2 ·

Basic reporting

significantly improved

Experimental design

same; description is improved and clearer

Validity of the findings

no comment

Additional comments

The revised manuscript is a significant improvement. The authors have addressed my concerns and made the recommended changes.

---

## Round 0.3 · accepted · Accept

Thank you for the revised manuscript.

#